# Early evolution of the ecdysozoan body plan

Deng Wang[1†], Yaqin Qiang[2†], Junfeng Guo[2]*, Jean Vannier[3†], Zuchen Song[2], Jiaxin Peng[2], Boyao Zhang[2], Jie Sun[1,2], Yilun Yu[4,5], Yiheng Zhang[6], Tao Zhang[6], Xiaoguang Yang[1], Jian Han[1]*

[1]State Key Laboratory of Continental Dynamics, Shaanxi Key Laboratory of Early Life & Environments and Department of Geology, Northwest University, Xi'an, China; [2]School of Earth Science and Resources, Key Laboratory of Western China's Mineral Resources and Geological Engineering, Ministry of Education, Chang'an University, Xi'an, China; [3]Université de Lyon, Université Claude Bernard Lyon 1, ENS de Lyon, CNRS, Laboratoire de Géologie de Lyon: Terre, Planètes, Environnement (CNRS-UMR 5276), Villeurbanne, France; [4]Institute of Vertebrate Paleontology and Paleoanthropology, Chinese Academy of Sciences, Beijing, China; [5]University of Chinese Academy of Sciences, Beijing, China; [6]School of Information Science and Technology, Northwest University, Xi'an, China

*For correspondence:
junfengg@chd.edu.cn (JG);
elihanj@nwu.edu.cn (JH)

[†]These authors contributed equally to this work

Competing interest: The authors declare that no competing interests exist.

**Abstract** Extant ecdysozoans (moulting animals) are represented by a great variety of soft-bodied or articulated organisms that may or may not have appendages. However, controversies remain about the vermiform nature (i.e. elongated and tubular) of their ancestral body plan. We describe here *Beretella spinosa* gen. et sp. nov. a tiny (maximal length 3 mm) ecdysozoan from the lowermost Cambrian, Yanjiahe Formation, South China, characterized by an unusual sack-like appearance, single opening, and spiny ornament. *Beretella spinosa* gen. et sp. nov has no equivalent among animals, except *Saccorhytus coronarius*, also from the basal Cambrian. Phylogenetic analyses resolve both fossil species as a sister group (Saccorhytida) to all known Ecdysozoa, thus suggesting that ancestral ecdysozoans may have been non-vermiform animals. Saccorhytids are likely to represent an early off-shot along the stem-line Ecdysozoa. Although it became extinct during the Cambrian, this animal lineage provides precious insight into the early evolution of Ecdysozoa and the nature of the earliest representatives of the group.

## eLife assessment

This study provides a **fundamental** advance in palaeontology by reporting the fossils of a new invertebrate, Beretella spinosa, and inferring its relationship with already described species. The analysis placed the newly described species in the earliest branch of moulting invertebrates. The study, supported by **convincing** fossil observation, hypothesizes that early moulting invertebrate animals were not vermiform.

## Introduction

The Ediacaran–Cambrian transition is marked by the appearance in the fossil record of a variety of new body plans that prefigure the majority of present-day animal lineages, including the ecdysozoans, a huge clade that encompasses all invertebrate animals growing through successive moulting stages, such as panarthropods (Arthropoda, Onychophora, Tardigrada), scalidophoran (incl. Priapulida) and nematoid worms (*Erwin, 2020*). Altogether ecdysozoans represent a very high percentage

of animal biodiversity and disparity, inhabiting almost all possible ecological niches on Earth (*Brusca et al., 2016*). The nature of the last common ancestor of Ecdysozoa (LCAE) remains largely unresolved, even though worms are prevalent before the rise of panarthropods as trace and body fossils in basal Cambrian and late Ediacaran rocks (*Buatois et al., 2014*; *Liu et al., 2014*; *Vannier et al., 2010*). Some recent molecular phylogenies also predict that the most basal ecdysozoans were worm-like, elongated organisms (*Howard et al., 2022*; *Laumer et al., 2019*) that possibly diverged in the Ediacaran (*Howard et al., 2022*; *Rota-Stabelli et al., 2013*). Current reconstruction based on fossil and developmental evidence features the ancestral ecdysozoan as a millimeter-sized worm (*Budd, 2001*; *Valentine and Collins, 2000*) with a terminal (*Ortega-Hernández et al., 2019*) or ventral mouth (*Martín-Durán and Hejnol, 2015*; *Nielsen, 2019*). Clearly, the discovery of *Saccorhytus* (*Han et al., 2017*; *Liu et al., 2022*; *Shu and Han, 2020*) in the basal Cambrian of China (Kuanchuanpu Formation; ca. 535 Ma *Sawaki et al., 2008*) that is anything but a worm sowed doubt among scientists. *Saccorhytus* is a sac-like secondarily phosphatized microscopic animal spiked with conical sclerites and a single opening that was first seen as the earliest known deuterostome (*Han et al., 2017*) but is now considered as an ecdysozoan on more solid grounds (*Liu et al., 2022*; *Shu and Han, 2020*), thus broadening the anatomical spectrum of the group and its disparity in the Cambrian and reopening the debate on the nature of LCAE.

We describe here *Beretella spinosa* gen. et sp. nov. from Member 5 of the Yanjiahe Formation (basal Cambrian Stage 2, ca. 529 Ma, Hubei Province, China) that shares morphological traits with *Saccorhytus coronarius* such as an ellipsoidal body, a pronounced bilaterality, a spiny ornament made of broad-based sclerites, and a single opening. Cladistic analyses are made to resolve the position of both *Beretella* and *Saccorhytus* that provide key information on the early evolution of the group.

## Results
### Systematic palaeontology

> Superphylum Ecdysozoa *Aguinaldo et al., 1997*
> Phylum Saccorhytida Han, Shu, Ou and Conway Morris, 2017 stat. nov.

### Remarks
Saccorhytida first appeared in the literature as a new stem-group Deuterostomia that accommodated a single species, *Saccorhytus coronarius* (*Han et al., 2017*). Since *Saccorhytus* is no longer considered a primitive deuterostome and, instead, more likely belongs to ecdysozoans, Saccorhytida became an extinct Order of Ecdysozoa (*Liu et al., 2022*; *Shu and Han, 2020*). Because both *Saccorhytus* and *Beretella* display major morphological differences with all other known ecdysozoan phyla (Nematoida, Scalidophora, and Panarthropoda), Saccorhytida is tentatively elevated here to the rank of phylum within Ecdysozoa.

### Emended diagnosis
Microscopic, ellipsoidal body shape with pronounced bilateral symmetry expressed by paired spiny sclerites. Single, presumably oral opening on assumed ventral side (no anus).

### Remarks
Only two forms, *Saccorhytus* and *Beretella* are currently placed within Saccorhytida, making it premature to formally define intermediate taxonomic categories such as an order and a family.

> *Beretella spinosa* Han, Guo, Wang and Qiang, gen. et sp. nov.
> LSID: urn:lsid:zoobank.org:act:C2DC9EC2-82EB-4B2B-9829-718EE8104593

### Etymology
From '*béret*', French, that designates a soft, visorless cap referring to the overall shape of this species, and '*spinosa*', Latin, an adjective, alluding to its spiny ornament.

## Holotype
CUBar138-12 (*Figure 1A–C*, *Figure 1—figure supplement 1G*).

## Paratype
CUBar171-5 (*Figure 1H, I*, *Figure 1—figure supplement 1F*) and CURBar121-8 (*Figure 1J and K*, *Figure 1—figure supplement 1H*).

## Diagnosis
Body with a beret-like lateral profile. Convex side (presumably dorsal) with an elevated (presumably posterior) and lower (presumably anterior) end. The opposite side (presumably ventral) flattened. Bilateral symmetry well expressed in the overall body shape (sagittal plane) and sclerite distribution. Antero-posterior polarity. Convex side with a slightly elevated sagittal stripe topped with a single row of four aligned spines (S1) and five additional spines (S2) on each side. Six broad-based conical sclerites (S3) distributed in two symmetrical longitudinal rows plus two sagittal ones. Double rows of six marginal spines (S4 and S5). Flattened side often pushed in and partly missing, bearing a possible mouth opening. Possible oral spine.

## Stratigraphy and locality
*Watsonella crosbyi* Assemblage Zone (*Guo et al., 2021*), Member 5 of the Yanjiahe Formation (Cambrian Terreneuvian, Stage 2) in the Yanjiahe section near Yichang City, Hubei Province, China (*Figure 1—figure supplements 2 and 3*).

## Description and comparisons
The body of *Beretella spinosa* is secondarily phosphatized and has a consistent beret-like three-dimensional shape in the lateral view. Its length, width, and height range from 1000–2900 µm, 975–2450 µm, and 500–1000 µm, respectively (*Figure 1—figure supplement 1E-H*, *Figure 2—figure supplement 1E-I*, *Supplementary file 1a-c*). The ratio of the maximal length to width is 1.6:1 (*Figure 1—figure supplement 1E-H*). As seen in top view, *B. spinosa* shows a small lateral constriction at approximately mid-length (*Figure 1A and C*).

The body has a convex, assumedly dorsal side with one, presumably posterior end more elevated than the other (*Figure 1B, E, I and K*, *Figure 1—figure supplement 1E-H*, *Figure 2—figure supplement 1G-I*). This elevation is gradual along the sagittal plane and then becomes more abrupt near the low elevated, presumably anterior end. The opposite, assumedly ventral side is less well preserved and seems to have been originally flattened.

The convex side bears a complex ornamented pattern made of five sets (S1–S5) of spiny sclerites directed towards the more elevated end (*Figures 1A, B, D, E, H–K and 2A, B, D*, *Figure 2—figure supplement 1E, F*). These sclerites were originally pointed (*Figure 1A, B, D, E, H-K* and *Figure 2B, K, L*, *Figure 2—figure supplement 1A, B, G*), but most of them were broken thus revealing an internal cavity and an ellipsoidal transverse section (*Figures 1A, B, H–K and 2A-E, G*). The broken sclerites show an inner and outer phosphatic layer (thickness ca. 20–50 µm) often separated by a thin empty space (*Figure 2G-L*).

The convex side bears six prominent conical sclerites (S3) all with a rounded to elliptical well-delimited broad base, distributed in two longitudinal symmetrical pairs with two additional sclerites at both ends of the sagittal plane (*Figures 1 and 2D*, *Figure 1—figure supplement 1E-H*, *Figure 2—figure supplement 1E-I*). A low-relief stripe runs in a sagittal position and vanishes towards the elevated end. It is topped by a row of aligned spines (S1, *Figure 1A*); the one closer to the more elevated end being more tubular and longer. This row is flanked on both sides by smaller aligned spines (S2, *Figures 1A, D, H , and 2A–C*). Two relatively sinuous rows of six tiny spines are present parallel to the lateral margins (S4 and S5, *Figures 1B, E, H–J and 2D, E*).

The convex side bears a polygonal micro-ornament (mesh size ca. 5 µm wide, *Figure 2F*, *Supplementary file 1a-c*). However, its exact extension is uncertain due to coarse secondary phosphatization. Clusters of spherical phosphatized grains (diameter ca. 20 µm) occur near the sclerite base (*Figure 2—figure supplement 1B*).

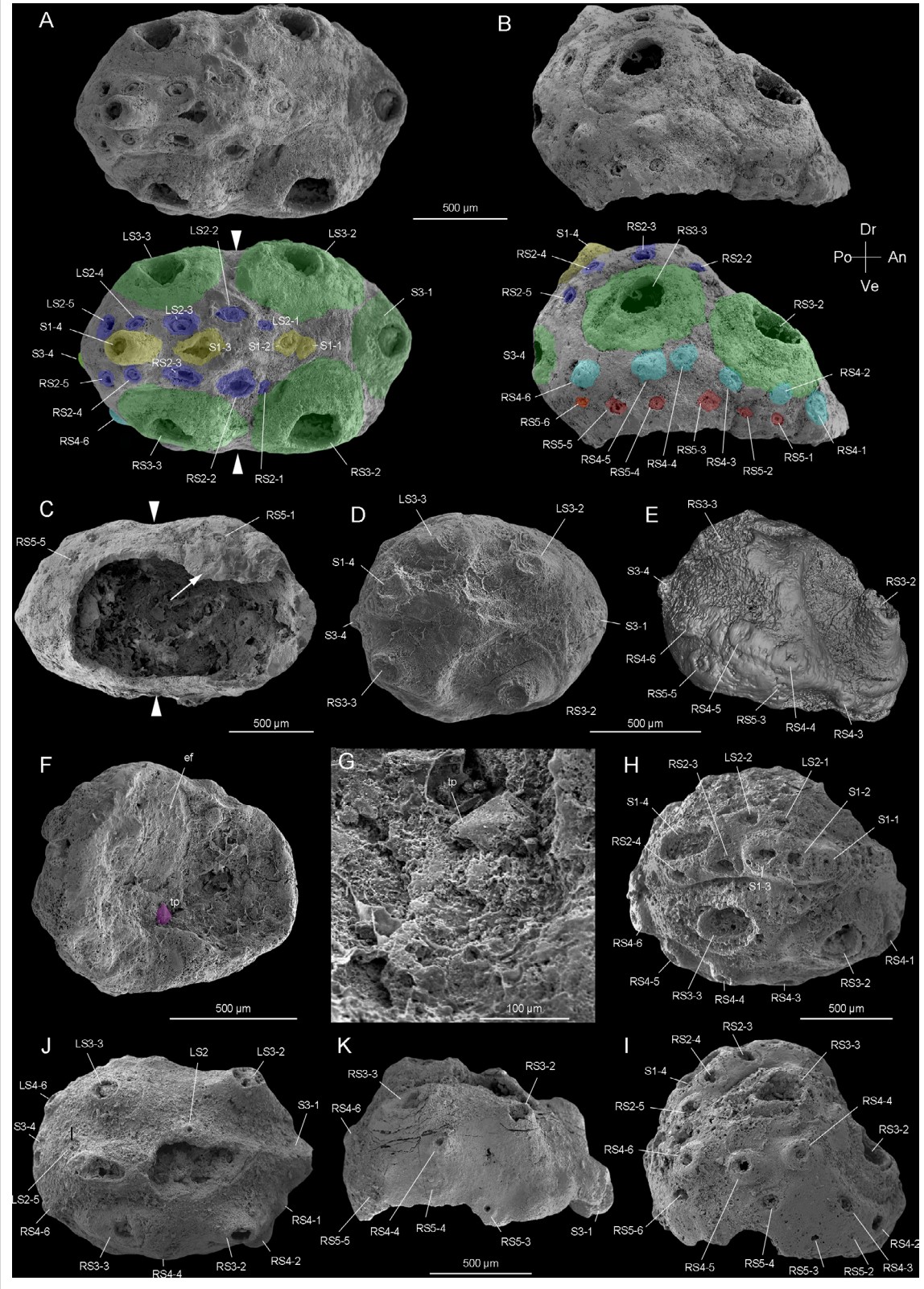

**Figure 1.** *Beretella spinosa* gen. et sp. nov. from Member 5 of the Yanjiahe Formation (Cambrian Stage 2), Yichang, Hubei Province, China. (**A–C**) Holotype, CUBar138-12. (**A**) Dorsal view showing the external ornament: (five sclerites at the midline in yellow (S1); flanked by two rows of sclerites in blue (S2); large broad-based conical sclerites in two dorsolateral pairs and one antero-posterior pairs in green (S3)); white arrows indicate lateral constriction. (**B**) Right lateral view showing two additional rows of six sclerites (S4 and S5, in light blue and pink, respectively). (**C**) Ventral view showing a

*Figure 1 continued on next page*

*Figure 1 continued*

large opening that may have accommodated the mouth (see the text) and an empty body cavity. (**D–G**) CUBar75-45. (**D**) Dorsal view showing a broken S3. (**E**) Micro-CT image, right lateral view displaying S4. (**F**) Ventral view depicting a tiny projection in purple. (**G**) An enlargement of the projection of F. (**H–I**) Paratype, CUBar171-5. (**H**) Right dorsal view showing S1–S4. (**I**) Right-lateral view showing S4 and S5. (**J–K**) Paratype CUBar121-8. (**J**) Dorsal view showing poorly preserved S1 and S2. (**K**) Right-lateral view showing S3–S5. A, assumed anterior end (see text); ef, exotic fragment; D, assumed dorsal side; L, left; P, posterior end; R, right; tp, tiny spine; V, ventral side. The same abbreviations are used throughout the manuscript including supplementary files.

The online version of this article includes the following figure supplement(s) for figure 1:

**Figure supplement 1.** Size variation between *Saccorhytus coronarius* and *Beretella spinosa*.

**Figure supplement 2.** Origin of fossil material.

**Figure supplement 3.** Typical Small Shelly Fossils (SSFs) found associated with *Beretella spinosa* in Member 5 of the Yanjiahe Formation.

In most specimens, the flattened side is occupied by a relatively large opening (1200 and 600 µm in maximal length and width, respectively) with irregularly defined margins (*Figure 1C and F*, see also *Video 1* and *Video 2*). The flattened side is often largely missing and opens into a spacious internal cavity with no signs of internal organs (e.g. gut and pharynx; *Figure 1C and F*). One specimen shows a tiny spine on the margin of the flattened side (*Figure 1F and G*), which differs from other spiny sclerites (S1-S5).

The length of studied specimens ranges from 1.0 to 2.9 mm (*Figure 1—figure supplement 1E-H*). Whether growth was continuous or instead took place via successive moulting stages and cuticular renewal (ecdysis) could not be tested due to the small number of specimens (N=17) available for measurements. No major morphological variations (e.g. a sclerite pattern) can be seen between the smallest and largest specimens of *B. spinosa* (*Figure 1—figure supplement 1E-H*).

## Remarks

### Body polarities in *Beretella*

The anterior-posterior (AP) and dorsal-ventral (DV) polarities of *Beretella* are uneasy to define because of the lack of modern equivalent among extant animals. In the vast majority of extinct and extant invertebrates for which antero-posterior polarity is defined on the basis of independent criteria (e.g. position of the mouth), sclerites point backwards (e.g. Cambrian scalidophoran worms [*Han et al., 2007*; *Huang et al., 2004*] and *Wiwaxia* [*Zhang et al., 2015a*]). This is most probably also the case with *Beretella* (*Figure 1A, D and J*). The dorsoventral polarity of *Beretella* is supported by the fact that protective sclerites such as spines most commonly occur on the dorsal side of bilaterians (*Figure 1A, D and J*).

### Comparison with *Saccorhytus* and other ecdysozoans

*Beretella spinosa* has no exact equivalent in any Cambrian animals except *Saccorhytus coronarius*, an enigmatic, sac-like ecdysozoan (*Han et al., 2017*; *Liu et al., 2022*; *Shu and Han, 2020*). Both forms share a tiny, poorly differentiated ellipsoidal body, and a set of prominent bilaterally arranged spiny sclerites. Indeed, the broad-based conical sclerites (S3) of *Beretella* are almost identical to those of *Saccorhytus* (*Figure 2—figure supplement 1D*) and have counterparts among scalidophoran worms (*Figure 2—figure supplement 1C*). However, they differ in number, ornamented structures, shape, and spatial arrangement (*Figure 2—figure supplement 2*). *Beretella* has a much more pronounced dorsoventral differentiation than *Saccorhytus* and its cuticle seems to have been harder and less flexible (see details in *Supplementary file 1a-c*), which altogether the hypothesis of *Saccorhytus* being the larval stage of *Beretella* unlikely. Both *Beretella* and *Saccorhytus* differ from other known ecdysozoans in the lack of an elongated body, introvert, annulations, and through gut (*Figure 1—figure supplement 1*, *Figure 2—figure supplements 1 and 2*, *Supplementary file 1a-c*).

## Discussion

### Ventral mouth

All bilaterian animals have a digestive system with at least one opening that corresponds to the mouth (*Brusca et al., 2016*). Although the presumed oral area of *Beretella* is poorly preserved (ventral side

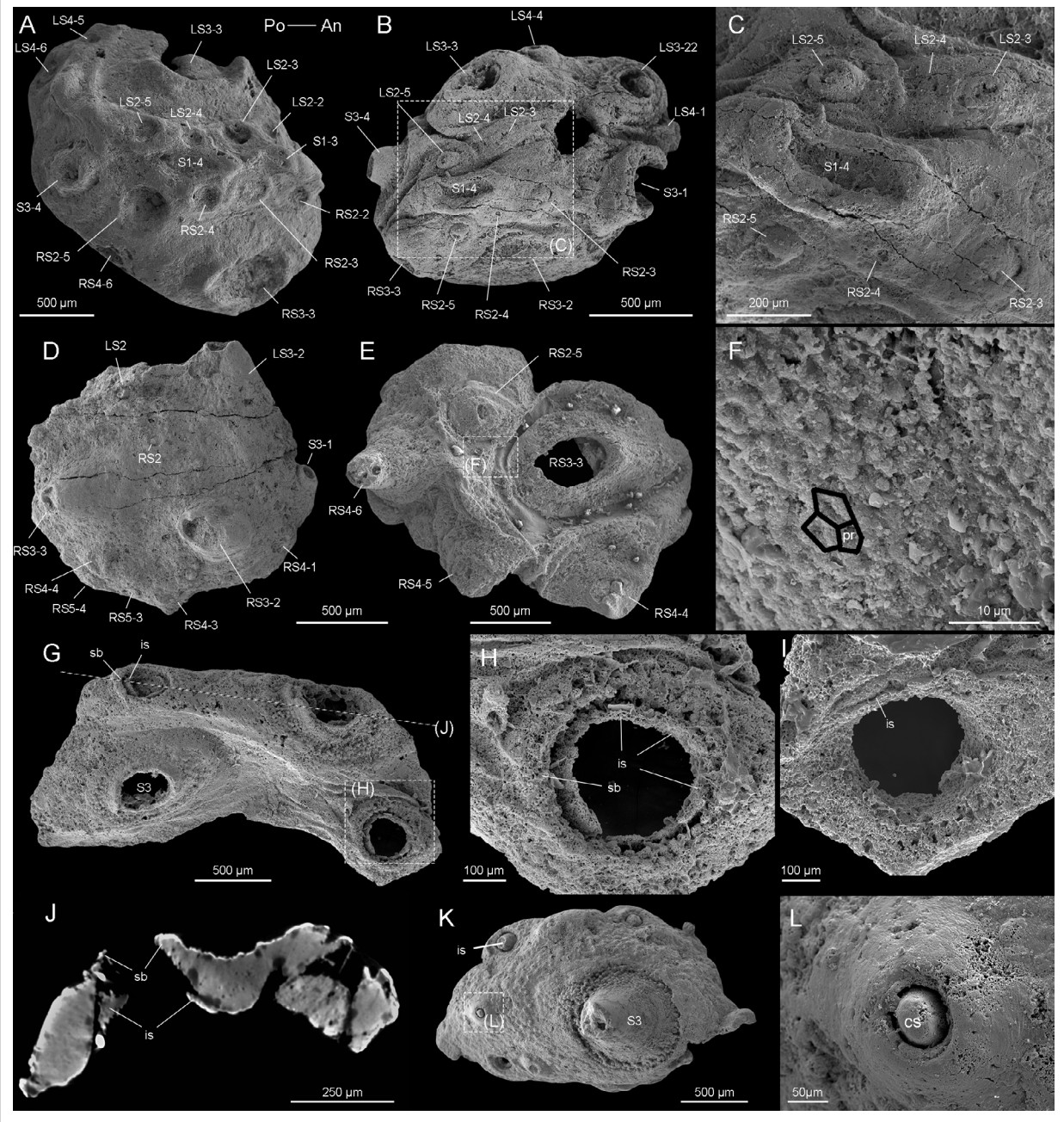

**Figure 2.** *Beretella spinosa* gen. et sp. nov. (**A**) CUBar99-19, dorsal view showing an ornament S1–S4. (**B, C**) CUBar136-9, general dorsal view and details. (**D**) CUBar136-11, dorsal view showing S1–S5. (**E, F**) CUBar73-15 general view and details of the cuticular polygonal reticulation in black. (**G–J**) CUBar128-27. (**G**) General view. (**H, I**) details of outer and inner surface of the bi-layered structure of the cuticular wall as seen in broken conical sclerites. (**J**) Micro-CT section showing possibly sclerite infilling. (**K, L**) CUBar99-18, cuticular fragment, general view and details of large sclerite (central feature represents possible phosphatic infilling). is, infilling sclerite; pr, polygonal reticulation; sb, sclerite base.

The online version of this article includes the following figure supplement(s) for figure 2:

**Figure supplement 1.** Truncated sclerites in early Cambrian saccorhytids and scalidophoran worms.

**Figure supplement 2.** *Saccorhytus coronarius*, multi-layered secondarily phosphatized cuticle.

often pushed in and largely destroyed), its mouth is likely to be found ventrally (see description), since no other opening occurs on its dorsal side, except those created by broken sclerites. The well-defined dorsoventral polarity of *Beretella* would suggest that the animal was resting on its ventral (flattened) side, the spiny dorsal side playing a protective role.

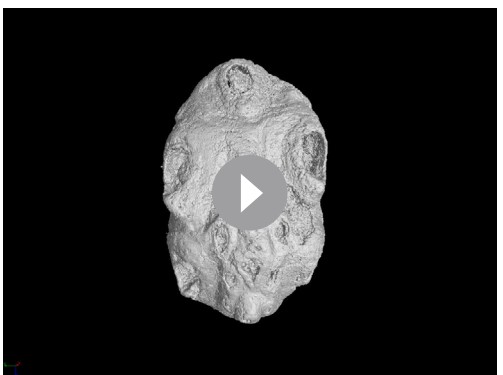

**Video 1.** Animation of holotype of *Beretella spinosa*.
https://elifesciences.org/articles/94709/figures#video1

## Phylogenetic position of *Beretella*

*Beretella*'s phylogenetic affinities remain elusive due to the lack of information concerning its internal anatomy and ventral side. Its scleritome consists of isolated conical sclerites that were the cuticular outgrowths of a seemingly rigid integument that covered both sides of the animal. Such conical sclerites have close counterparts in Cambrian ecdysozoans such as scalidophoran worms (e.g. *Eokinorhynchus Zhang et al., 2015b*), lobopodians (e.g. *Onychodictyon ferox Hou et al., 1991*) and even more clearly *Saccorhytus* that recent cladistic analyses resolved as a branch of the total-group Ecdysozoa (*Liu et al., 2022*). These sclerites, unknown in other animal groups, suggest that both *Saccorhytus* and *Beretella* belongs to Ecdysozoa (moulting animals), although more direct fossil evidence such as exuviae or features suggesting cuticular moulting (*Daley and Drage, 2016*; *Wang et al., 2019*) has yet to be found.

Cladistic analyses were performed to test the relation of *Beretella* and *Saccorhytus* to other ecdysozoan groups and, more generally, their phylogenetic relationships with other bilaterian groups (see details in *Figure 3—figure supplements 1–4*). Both taxa join in a clade (Saccorhytida, *Figure 3A–C*) and are resolved as members of total-group Ecdysozoa. This clade is the sister group of Cycloneuralia plus Panarthropoda (crown-group Ecdysozoa, *Figure 3D*, *Figure 3—figure supplements 1–4*, *Figure 4*). These results are consistent with the body plan of Saccorhytida being markedly different from that of crown-group ecdysozoans that all have an elongated body and differentiated structures, such as, in Cycloneuralia, the introvert and pharyngeal complex (*Figure 4*).

## The ancestral ecdysozoan body plan

Molecular clock analyses often place the divergence of Ecdysozoa relatively deep into the Ediacaran (*Howard et al., 2022*; *Rota-Stabelli et al., 2013*), thus highlighting major discrepancy with the known fossil record of the group. Potential ecdysozoans occur in the late Precambrian as suggested by sclerites resembling scalids of priapulids, found in Ediacaran Small Carbonaceous Fossils assemblages (*Moczydłowska et al., 2015*) and locomotion traces presumably made by scalidophoran worms (*Buatois et al., 2014*; *Vannier et al., 2010*). In the absence of fossil data for other groups such as nematoids, scalidophorans are potentially the oldest known representatives of Ecdysozoa. Recent Bayesian analyses based on a large molecular data set obtained from the 8 extant ecdysozoan phyla recover Scalidophora as the sister-group to Nematoida +Panarthropoda and suggest that ecdysozoans probably diverged in the Ediacaran, possibly some 23 million years before the oldest fossil occurrence (trace fossils) of the group (*Howard et al., 2022*). Although this study does not speculate on the nature of the last common ancestor of Ecdysozoa, it is consistent with the view that the earliest representatives of the group were probably worm-like, relatively elongated animals. (*Howard et al., 2020*) drew comparable conclusions based on *Acosmia*, an assumed stem-ecdysozoan worm from early Cambrian Chengjiang Lagerstätte. However, the re-evaluation of the morphological characteristics of this worm rather suggests a less basal position either within the total-group Cycloneuralia (*Figure 3D*, *Figure 3—figure supplements 1 and 2*) or among crown-group Ecdysozoa (*Figure 3—figure supplements 3 and 4*). The ellipsoidal (non-vermiform) shape of saccorhytids

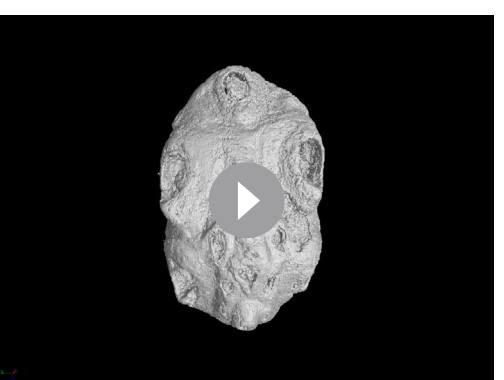

**Video 2.** Animation of holotype of *Beretella spinosa* without color.
https://elifesciences.org/articles/94709/figures#video2

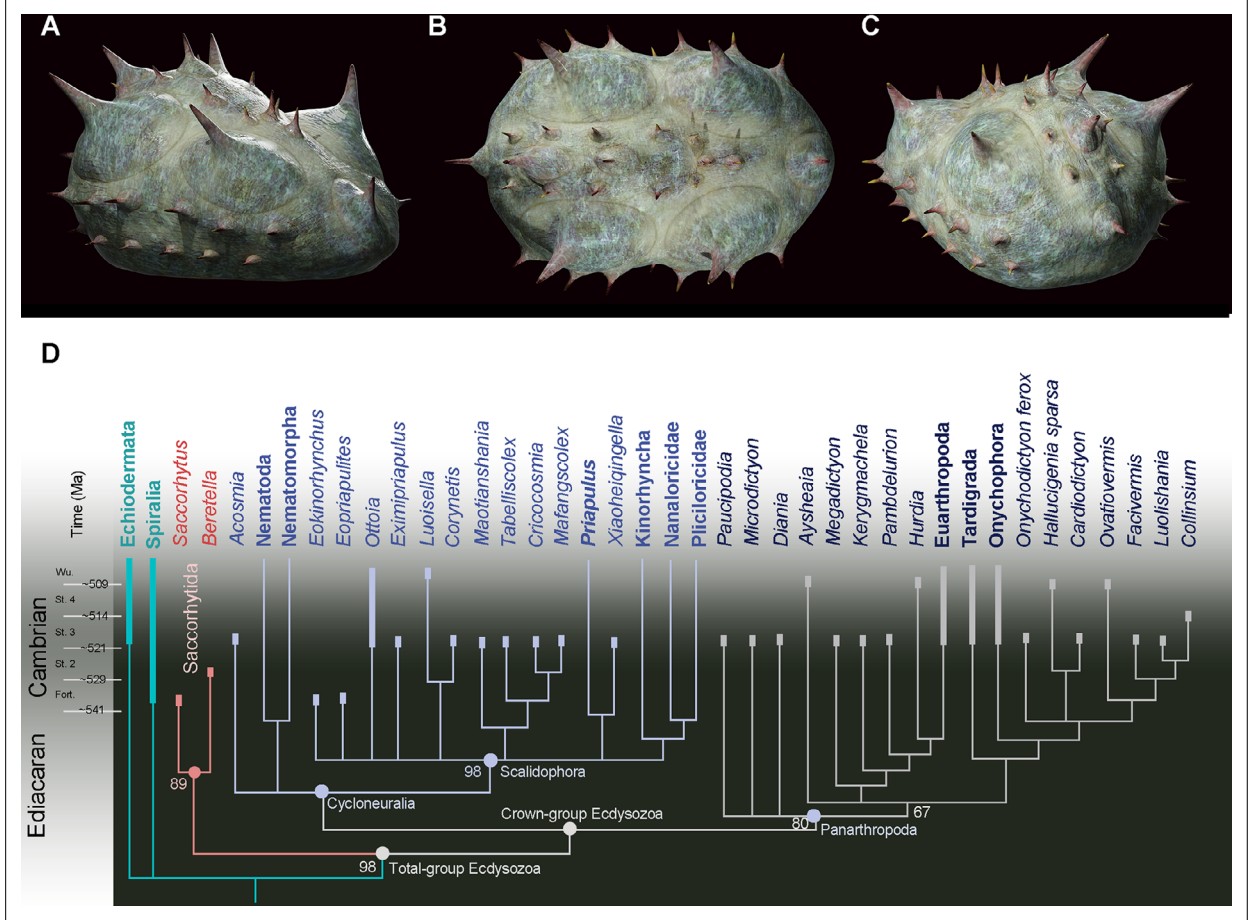

**Figure 3.** Position of *Beretella spinosa* in the animal tree based on cladistic analysis. (**A-C**), artistic three-dimensional reconstructions of *Beretella spinosa* in the anterolateral (**A**), dorsal (**B**), and posterolateral views (**C**). (**D**) Phylogenetic tree obtained from cladistic analyses using maximum likelihood. *Saccorhytus* and *Beretella* join in a clade (new phylum Saccorhytida) resolved as the sister-group of all other ecdysozoans; numbers at key nodes denote probability. Fossil and extant taxa are in italics and bold, respectively. Known fossil record indicated by thicker vertical bars (after ***Shu and Han, 2020***).

The online version of this article includes the following source data, source code, and figure supplement(s) for figure 3:

**Source code 1.** The dataset (matrix) for cladistic analysis.

**Source data 1.** Characters description for cladistic analysis.

**Figure supplement 1.** Full maximum likelihood tree generated by IQTREE.

**Figure supplement 2.** Bayesian inference tree generated by MrBayes.

**Figure supplement 3.** Maximum parsimony tree generated by TNT (equal weight).

**Figure supplement 4.** Maximum parsimony tree generated by TNT (implied weight, k=3).

and their position as the sister group of the crown-group Ecdysozoa clearly reopens the debate on the nature of the ancestral ecdysozoan (***Figure 4***) and has led to explore alternative evolutionary hypotheses, in particular: (i) does the enigmatic saccorhytid body plan results from anatomical simplification? (ii) to what extent may these animals shed light on the nature of the earliest ecdysozoans?

## Do saccorhytids result from simplification?

A relatively simple body plan and tiny size is often seen as resulting from anatomical simplification (e.g. reduction of digestive system) and miniaturization (micrometric size) in possible relation with the adaptation to specialized ecological niches or parasitism (***Hanken and Wake, 1993***). For example, some extant scalidophoran worms living in interstitial (meiobenthic) habitats such as loriciferans have a miniaturized body (***Kristensen, 1983***) compared with their macroscopic counterparts (e.g. *Priapulus* ***Schmidt-Rhaesa, 2013b***). However, they retain a through gut and a functional introvert and show

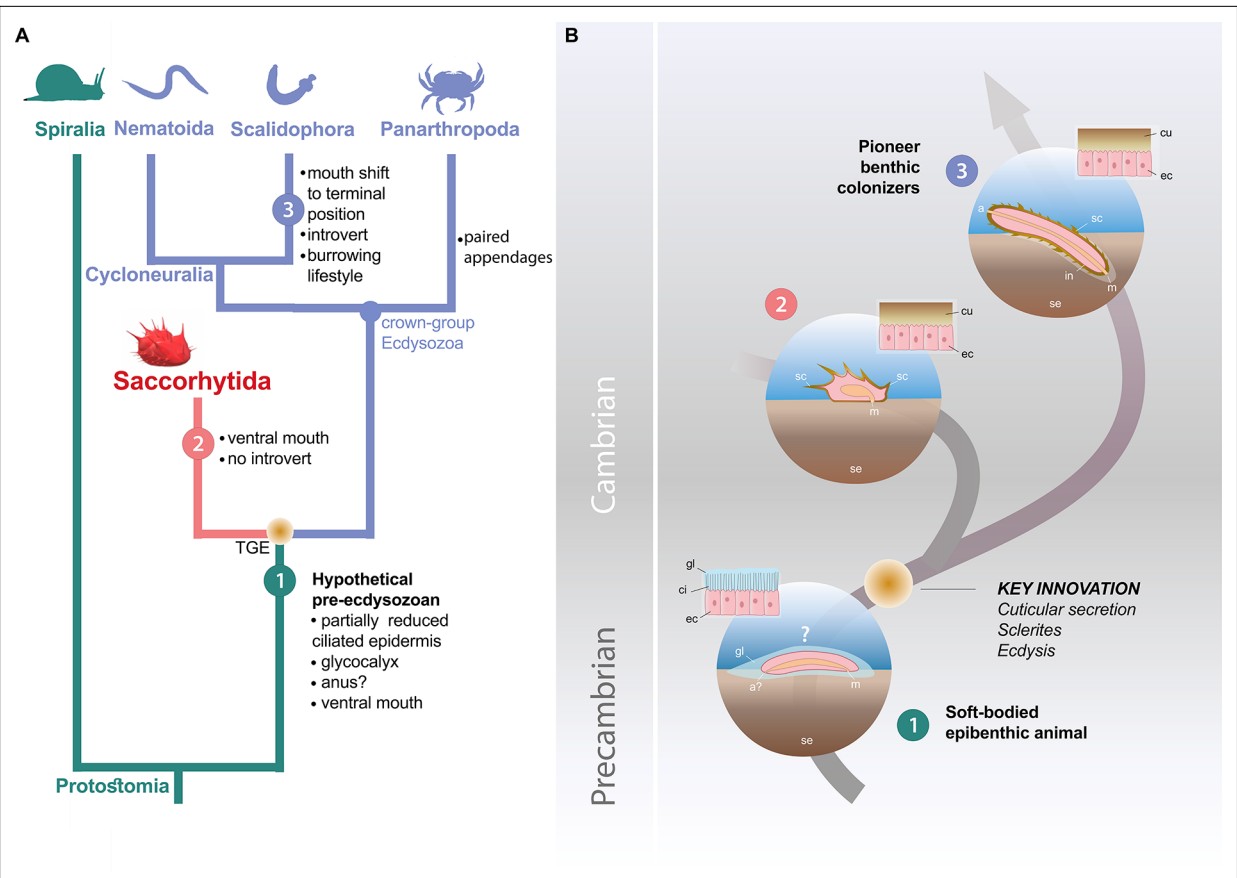

**Figure 4.** Possible evolutionary scenario to explain the origin and early evolution of ecdysozoans. (**A**) Summary tree (see *Figure 3—figure supplements 1–4*) showing saccorhytids as a sister-group of Cycloneuralia (Nematoida plus Scalidophora)+Panarthropoda; main morphological features of each group listed along each branch. (**B**) Potential evolutionary pathway to evolve Saccorhytida and crown-group Ecdysozoa. Numbers in green, red and blue circles designate pre-ecdysozoan (Spiralia), Saccorhytida and Cycloneuralia, respectively. Light brown gradient (circle) to emphasize ecdysis and sclerite secretion seen as key evolutionary steps. 1, Hypothetical pre-ecdysozoan animal with a ciliated epidermis and glycocalyx. 2, Saccorhytid exemplified by *Beretella* with a cuticle bearing sclerites. 3, Crown-group ecdysozoan exemplified by a scalidophoran worm with an elongated shape, a differentiated head (introvert) and trunk, sclerites, a through gut, a terminal mouth and abilities to burrow into bottom sediment. Animals not to scale. Abbreviations: a, anus; a?, uncertain status of anus; ci, cilia; cu, cuticle; ec, epidermal cell; gl, glycocalyx (mucous layer); m, mouth; in, introvert; sc, sclerite; se, sediment; TGE, total-group Ecdysozoa. Silhouettes from phylopic.org. (CC BY 3.0 or public domain): Spiralia (by Martin R. Smith), Nematoida (by Birgit Lang), Scalidophora (by Fernando Carezzano), and Panarthropoda (by Harold N Eyster). Saccorhytida generated from reconstruction of *Figure 3*.

no sign of drastic internal simplification (*Schmidt-Rhaesa, 2013a*). Anatomical reduction is a typical feature of parasitism (*Hanken and Wake, 1993*) that is well-represented among extant ecdysozoans such as nematodes (*Schmidt-Rhaesa, 2013c*). Although relatively small (ca. 0.1–2.5 mm long), nematodes underwent no simplification of their digestive system. Saccorhytids have no specialized features (e.g. anchoring or piercing structures) that would point to any adaptation to ecto- or endo-parasitic lifestyles (*Cong et al., 2017*). *Saccorhytus* has been interpreted (*Han et al., 2017*) as a possible interstitial animal based on its micrometric size which corresponds to that of the extant meiofauna. If we accept the hypothesis that saccorhytids result from simplification, then we need to determine its origin. Simplification of saccorhytids from an elongated animal (e.g. cycloneuralian worm with a through gut and terminal mouth) is difficult to conceive because it would involve considerable anatomical transformations such as the loss of tubular organization, introvert and pharynx in addition to that of the digestive system (*Figure 4*, and *Supplementary file 1d, e*).

## Early evolution of ecdysozoans: a new scenario

We propose here an alternative evolutionary hypothesis (*Figure 4*) in which saccorhytids are replaced within the broader framework of the origin and early diversification of moulting animals. Saccorhytids

are seen as an early off-shot from the stem-line Ecdysozoa (see cladistic analysis above) that possibly retained important features of the body plan of ancestral ecdysozoans. This scenario must be considered as a working hypothesis whose aim is to stimulate research in this key area of animal evolution.

The cuticular secretion and the loss of cilia (*Valentine and Collins, 2000*) would be the first of a series of evolutionary events (*Figure 4*) that led to the rise of Ecdysozoa. Moulting (shedding of the old cuticle via apolysis and its renewal) reconciled body growth and cuticular protection (*Schmidt-Rhaesa, 2007*). Cuticle secretion and moulting may have been quasi-simultaneous innovations that took place over a relatively short time interval. The nature of the very first ecdysozoans is hypothetical and lacks fossil evidence. However, they are tentatively represented here as small epibenthic or interstitial slow-moving non- elongated animals from which saccorhytids may have evolved.

In our scenario, this ancestral ecdysozoan stock would have also given rise to elongated and tubular ecdysozoans through stepwise anatomical transformations such as the body elongation, the differentiation of key morpho-functional structures such as the pharynx and the introvert and the shift of the ventral mouth to a terminal position (*Martín-Durán and Hejnol, 2015*; *Figure 4*, *Supplementary file 1d, e*). This mouth shift from ventral to terminal arising in crown ecdysozoans is consistent with the chronology of divergence of animal lineages and the fact that the mouth of most spiralians is ventral (*Martín-Durán and Hejnol, 2015*; *Nielsen, 2019*; *Ortega-Hernández et al., 2019*). Developmental studies show that embryos of extant cycloneuralians have a ventral mouth that moves to a terminal position towards the adult stage (*Martín-Durán and Hejnol, 2015*; *Nielsen, 2019*). These assumed major anatomical changes (e.g. functional introvert) must be placed in the ecological context of Cambrian animal radiation. Important changes in the functioning of marine ecosystems occurred in the early Cambrian such as interactive relationships between animal species, exemplified by predation (*Vannier and Chen, 2005*; *Vermeij, 1977*) and may have acted as drivers in the evolution of early ecdysozoans, in promoting burrowing into sediment and the colonization of endobenthic habitats for the first time (*Vannier et al., 2010*). Burrowing into the sediment could be seen as the evolutionary response of epibenthic animals such as ancestral ecdysozoans to escape visual predation (*Daley et al., 2013*; *Vannier and Chen, 2005*). We hypothesize that this migration to endobenthic shelters was made possible by the development of a resistant cuticular layer (*Figure 4*) that strongly reduced physical damage caused by friction with the sediment and provided anchoring points (e.g. scalids and sclerites). Whereas saccorhytids became rapidly extinct during the Cambrian, worms massively colonized endobenthic habitats, resulting in bioturbation and ecological turnover.

## Methods

### Material

Fourteen specimens of *Beretella spinosa* were recovered from samples (siliceous-phosphatic, intraclastic limestone) collected from Member 5 of the Yanjiahe Formation, Yanjiahe section near Yichang City, Hubei Province, China (*Guo et al., 2021*). These were obtained by digesting the rocks in 10% acetic acid. Faunal elements associated with *Beretella spinosa* in residues are mainly tiny molluscs (CUBar21-4 and CUBar206-6) (*Figure 1—figure supplement 3*). Comparisons were made with 10 specimens of *Saccorhytus coronarius* (ELIXX25-62, ELIXX34-298, ELIXX45-20, ELIXX48-64, ELIXX58-336, ELIXX61-27, ELIXX65-116, ELIXX65-296, ELIXX99-420) and one coeval scalidophoran specimen (ELIXX57-320) all from Bed 2 of the Kuanchuanpu Formation, Zhangjiagou section near Xixiang County, south Shaanxi Province, China. All specimens of *Beretella* are deposited in the paleontological collections of Chang'an University, Xi'an (CUBar), those of scalidophoran, and *Saccorhytus* at Northwest University, Xi'an (ELIXX), China.

### Scanning electron microscopy (SEM)

All specimens were coated with gold and then imaged using a FEI Quanta 400 FEG SEM at Northwest University and a FEI Quanta 650 at Chang'an University.

### X-ray computed microtomography and 3D reconstruction

Micro-CT-images (tiff format, with pixel size 1.1 μm) of *Beretella* (CUBar75-45, CUBar128-27, CUBar138-12) and *Saccorhytus* (ELIXX65-116, ELIXX99-420) were acquired using the Zeiss Xradia 520

at Northwest University (NWU), Xi'an, China, at an accelerating voltage of 50 kV and a beam current of 80 µA. Micro-CT data were processed using VGstudio Max 3.2 for 3D volume rendering.

## Measurements

Measurements of the length, width, and height of *Beretella* and *Saccorhytus* were obtained from Micro-CT and SEM images by using tipDig2 v.2.16.

## Phylogenetic analysis

We built our matrix with 55 taxa coded using 193 morphological characteristics (*Figure 3—source data 1*, *Figure 3—source code 1*). It is largely based on the data published by *Howard et al., 2020*, *Vinther and Parry, 2019* and *Ou et al., 2017*, although emended and supplemented by recent updates and new observations (*Figure 3—source data 1*, *Figure 3—source code 1*). Three characters (37. Through gut, 38. U-shaped gut, and 40. Ventral mouth) in matrix were coded as '? (uncertain)', '?', and '?', respectively. Because although we can infer a ventral mouth and no anus of *Beretella*, these anatomic structures are invisible in fossils. We analyzed the data matrix using maximum parsimony (Tree analysis using New Technology, TNT), maximum likelihood (Important quartet tree, IQTREE) and Bayesian inference (MrBayes). Parsimony analysis was implemented in TNT under equal and implied (k=3) weight. Parameters are default (*Goloboff et al., 2008*; *Goloboff and Catalano, 2016*). The maximum-likelihood tree search was conducted in IQ-TREE (*Nguyen et al., 2015*), and support was assessed using the ultrafast phylogenetic bootstrap replication method (*Hoang et al., 2018*; *Minh et al., 2013*) to run 50,000 replicates. Bayesian inference was conducted in with MrBayes v3.2.6a with default priors and Markov chain Monte Carlo settings (*Ronquist et al., 2012*). Two independent runs of 7,000,000 Markov chain Monte Carlo generations were performed, each containing four Markov chains under the Mkv + $\Gamma$ model for the discrete morphological character data (*Lewis, 2001*). In each run (N=2), trees were collected at a sampling frequency of every 5,000 generations and with the first 25% samples discarded as burn-in. The convergence of chains was checked by effective sample size (ESS) values over 1,000 in Tracer v.1.7 (*Rambaut et al., 2018*), 1.0 for the potential scale reduction factor (PSRF; *Gelman and Rubin, 1992*), and by an average standard deviation of split frequencies below 0.007.

## Ancestral character state reconstructions

Ancestral character state reconstructions for four morphological characters were performed on the ecdysozoan total group node, the ecdysozoan crown group node and saccorhytid node. Cycloneuralia was treated as (i) a monophyletic (*Supplementary file 1d*) and (ii) paraphyletic group (*Supplementary file 1e*). Characters selected for ancestral state reconstruction represent traits inferred as ecdysozoan plesiomorphies (ancestral characters) from studies of crown group taxa. These characters included the presence or absence of: (1) through gut; (2) ventral mouth; (3) introvert (see *Supplementary file 1d, e*).

This was carried out individually for the selected character in MrBayes. This was employed to calculate the posterior probability of the presence (1) and absence (0) of the selected characters at the selected nodes. Analyses used the MK +gamma model, and always converged after 2 million generations. Average deviation of split frequencies (<0.01), ESS scores (>200), and PSRF values (=approx. 1.00) assessed convergence of the MCMC chains (*Howard et al., 2020*).

## Acknowledgements

We thank H G for technical assistance. Funding: We thank the National Natural Science Foundation of China (grants 42172016, 41890844 to JG, 41621003, 42372012 to JH, 42202009 to DW), the Strategic Priority Research Program of the Chinese Academy of Sciences (grant XDB26000000 grant to JH and JG), the China Postdoctoral Science Foundation (grant 2022M722568 to DW), the Key Scientific and Technological Innovation Team Project in Shaanxi Province (grant to JG), National Key Research and Development Program of China (grant number 2023YFF0803601 to JH), 'open for collaboration' grant from Yunnan Key Laboratory for Palaeobiology, Yunnan University (to DW), and the Région Auvergne Rhône Alpes and Université Claude Bernard Lyon 1 (grant to JV) for financial support.

# Additional information

## Funding

| Funder | Grant reference number | Author |
| --- | --- | --- |
| National Key Research and Development Program of China | 2023YFF0803601 | Jian Han |
| National Natural Science Foundation of China | 42172016 | Junfeng Guo |
| National Natural Science Foundation of China | 41890844 | Junfeng Guo |
| National Natural Science Foundation of China | 41621003 | Jian Han |
| National Natural Science Foundation of China | 42372012 | Jian Han |
| National Natural Science Foundation of China | 42202009 | Deng Wang |
| Strategic Priority Research Program of the Chinese Academy of Sciences | XDB26000000 | Junfeng Guo Jian Han |
| China Postdoctoral Science Foundation | 2022M722568 | Deng Wang |
| Key Scientific and Technological Innovation Team Project in Shaanxi Province | | Junfeng Guo |
| Région Auvergne Rhône Alpes and Université Claude Bernard Lyon 1 | | Jean Vannier |
| Yunnan University | open for collaboration | Deng Wang |

The funders had no role in study design, data collection and interpretation, or the decision to submit the work for publication.

## Author contributions

Deng Wang, Conceptualization, Data curation, Software, Formal analysis, Funding acquisition, Investigation, Methodology, Writing – original draft, Writing – review and editing; Yaqin Qiang, Data curation, Visualization; Junfeng Guo, Conceptualization, Data curation, Funding acquisition, Project administration, Writing – review and editing; Jean Vannier, Formal analysis, Funding acquisition, Visualization, Writing – review and editing; Zuchen Song, Jiaxin Peng, Boyao Zhang, Data curation; Jie Sun, Yiheng Zhang, Software, Visualization, Methodology; Yilun Yu, Software, Formal analysis, Visualization, Methodology; Tao Zhang, Software; Xiaoguang Yang, Writing – review and editing; Jian Han, Conceptualization, Formal analysis, Supervision, Funding acquisition, Project administration, Writing – review and editing

## Author ORCIDs

Deng Wang https://orcid.org/0000-0002-4464-9632
Yaqin Qiang http://orcid.org/0000-0003-2967-0860
Junfeng Guo http://orcid.org/0000-0001-8743-5143
Jean Vannier http://orcid.org/0000-0003-0998-1231
Zuchen Song http://orcid.org/0000-0003-3786-5975
Jiaxin Peng http://orcid.org/0009-0006-0121-191X
Yiheng Zhang http://orcid.org/0009-0002-2426-0838
Tao Zhang http://orcid.org/0000-0002-5622-0227
Jian Han http://orcid.org/0000-0002-2134-4078

Reviewer #1 (Public Review): https://doi.org/10.7554/eLife.94709.3.sa1
Reviewer #2 (Public Review): https://doi.org/10.7554/eLife.94709.3.sa2
Author response https://doi.org/10.7554/eLife.94709.3.sa3

## Additional files

### Supplementary files

• Supplementary file 1. Meaurements of Saccorytida and ancestral character state reconstruction of Cycloneuralia. (a) Measurements of *Beretella*. L, length; W, width; H, height; ae, anterior end; B, body; pe, posterior end; PP, polygonal net-like pattern; tp, tiny spine; VO, ventral opening;?, no accurate measurement possible. (b) Length/width ratio of *Beretella* and *Saccorhytus*. (c) Similarities and differences between *Beretella* and *Saccorhytus*. AP, antero-posterior; DV, dorso-ventral side; LR, left-right. (d) Ancestral character state reconstructions for the topology where Cycloneuralia is monophyly. Values of ancestral character state reconstructions. 0=absence of character, 1=presence of character, *P*=posterior probability. TGE, total-group Ecdysozoa; CGE, crown-group Ecdysozoa, SA, Saccorhytida. (e) Ancestral character state reconstructions for the topology where Cycloneuralia is paraphyletic. Values of ancestral character state reconstructions. 0=absence of character, 1=presence of character, *P*=posterior probability. TGE, total-group Ecdysozoa; CGE, crown-group Ecdysozoa, SA, Saccorhytida.

• MDAR checklist

### Data availability

The data that support the findings of this study are available in the present paper and the supplementary files and source data.

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
