## [Editor Report · eLife assessment]

This study provides a **fundamental** advance in palaeontology by reporting the fossils of a new invertebrate, Beretella spinosa, and inferring its relationship with already described species. The analysis placed the newly described species in the earliest branch of moulting invertebrates. The study, supported by **convincing** fossil observation, hypothesizes that early moulting invertebrate animals were not vermiform.

---

## [Referee Report · Reviewer #1 (Public Review)]

Summary:

Wang and co-workers characterise the fossil of Beretella spinosa from the early Cambrian, Yanjiahe Formation, South China. Combining morphological analyses with phylogenetic reconstructions, the authors conclude that B. spinosa is closely related to Saccorhytus, an enigmatic fossil recently ascribed to Ecdysozoa, or moulting animals, as an extinct "basal" lineage. Finding additional representatives of the clade Saccorhytida strengthens the idea that there existed a diversity of body plans previously underappreciated in Ecdysozoa, which may have implications for our understanding of the earliest steps in the evolution of this major animal group.

Strengths:

I'm not a paleobiologist; therefore, I cannot give an expert opinion on the descriptions of the fossils. However, the similarities with Saccorhytus seem evident, and the phylogenetic reconstructions are adequate. Evolutionary interpretations are generally justified, and the consolidation of Saccorhytida as the extinct sister lineage to extant Ecdysozoans will have significant implications for our understanding of this major animal clade.

Weaknesses:

While I generally agree with the author's interpretations, the idea of Saccorhytida as a divergent, simplified off-shot is slightly contradictory with a probably non-vermiform ecdysozoan ancestor. The author's analyses do not discard the possibility of a vermiform ecdysozoan ancestor (importantly, Supp Table 4 does not reconstruct that character), and outgroup comparison with Spiralia (and even Deuterostomia for Protostomia as a whole) indicates that a more or less anteroposteriorly elongated (i.e., vermiform) body is likely common and ancestral to all major bilaterian groups, including Ecdysozoa. Indeed, Figure 4 b depicts the potential ancestor as a "worm". The authors argue that the simplification of Saccorhytida from a vermiform ancestor is unlikely "because it would involve considerable anatomical transformations such as the loss of vermiform organization, introvert and pharynx in addition to that of the digestive system". However, their data support the introvert as a specialisation of Scalidophora (Fig. 4a and Supp Table 4), and a pharyngeal structure cannot be ruled out in Saccorhytida. Likewise, loss of an anus is not uncommon in Bilateria. Moreover, this can easily become a semantics discussion (to what extent can an animal be defined as "vermiform"? Where is the limit?). Therefore, I suggest to leave the evolutionary scenario more open. Supporting Saccorhytida as a true group at the early steps of Ecdysozoa evolution is important and demonstrates that animal body plans are more plastic than previously appreciated. However, with the current data, it is unlikely that Saccorhytida represents the ancestral state for Ecdysozoa (as the authors admit), and a vermiform nature is not ruled out (and even likely) in this animal group. Suggesting that the ancestral Ecdysozoan might have been small and meiobenthic is perhaps more interesting and supported by the current data (phylogeny and outgroup comparison with Spiralia).

---

## [Referee Report · Reviewer #2 (Public Review)]

Summary:

This work provides important anatomical features of a new species from the Lower Cambrian, which helps advance our understanding of the evolutionary origins of animal body plans. The authors interpreted that the new species possessed a bilateral body covered with cuticular polygonal reticulation and a ventral mouth. Based on cladistic analyses using maximum likelihood, Bayesian, and parsimony, the new species was placed, along with Saccorhytus, in a sister-group ("Saccorhytida") of the Ecdysozoa. The phylogenetic position of Saccorhytida suggests a new scenario of the evolutionary origin of the crown ecdysozoan body plan.

Strengths:

Although the new species reported in this paper show strange morphologies, the interpretation of anatomical features was based on detailed observations of multiple fossil specimens, thereby convincing at the moment. Morphological data about fossil taxa in the Ediacaran and Early Cambrian are quite important for our understanding of the evolution of body plans (and origins of phyla) in paleontology and evolutionary developmental biology, and this paper represents a valuable contribution to such research fields.

Weaknesses:

The preservations of the specimens, in particular on the putative ventral side, are not good, and the interpretation of the anatomical features need to be tested with additional specimens in future. The monophyly of Cycloneuralia (Nematoida + Scalidophora) was not necessarily well-supported by cladistic analyses (Supplementary Figures 7-9), and the evolutionary scenario (Fig. 4) also need to be tested in future works. On the other hand, the revised version provides important contributions from currently available data, and the above-mentioned problems should be studied in a separate paper in future.

---

## [Author Response]

The following is the authors’ response to the original reviews.

**Public reviews:**

**Reviewer 1:**
Weaknesses:While I generally agree with the author's interpretations, the idea of Saccorhytida as a divergent, simplified off-shot is slightly contradictory with a probably non-vermiform ecdysozoan ancestor. The author's analyses do not discard the possibility of a vermiform ecdysozoan ancestor (importantly, Supplementary Table 4 does not reconstruct that character),

Saccorhytids are only known from the early Cambrian and their unique morphology has no equivalent among any extinct or extant ecdysozoan groups. This prompted us to consider them as a possible dead-end evolutionary off-shot. The nature of the last common ancestor of ecdysozoan (i.e. an elongated worm-like or non-vermiform animal with capacities to renew its cuticle by molting) remains hypothetical. At present, palaeontological data do not allow us to resolve this question. The animal in Fig. 4b at the base of the tree is supposed to represent an ancestral soft-bodied form with no cuticle from which ecdysozoan evolved via major innovations (cuticular secretion and ecdysis). Its shape is hypothetical as indicated by a question mark. Our evolutionary model is clearly intended to be tested by further studies and hopefully new fossil discoveries.

…and outgroup comparison with Spiralia (and even Deuterostomia for Protostomia as a whole) indicates that a more or less anteroposteriorly elongated (i.e., vermiform) body is likely common and ancestral to all major bilaterian groups, including Ecdysozoa. Indeed, Figure 4b depicts the potential ancestor as a "worm". The authors argue that the simplification of Saccorhytida from a vermiform ancestor is unlikely "because it would involve considerable anatomical transformations such as the loss of vermiform organization, introvert, and pharynx in addition to that of the digestive system". However, their data support the introvert as a specialisation of Scalidophora (Figure 4a and Supplementary Table 4), and a pharyngeal structure cannot be ruled out in Saccorhytida. Likewise, loss of an anus is not uncommon in Bilateria. Moreover, this can easily become a semantics discussion (to what extent can an animal be defined as "vermiform"? Where is the limit?).

We agree that “worm” and “vermiform” are ill-defined terms. They are widely used in various palaeontological and biological papers to describe elongated tubular animals such as edydsozoans and annelids (see Giribet and Edgecombe 2017; popular textbook written by Nielsen 2012; Schmit-Rhaesa 2013; Brusca et al. 2023; Giribet and Edgecombe 2020). Very few other animals are termed “worms”. Changes have been made in the text to solve this semantic problem, for example in the abstract where we added (i.e elongated and tubular) to better define what we mean by “vermiform”.

Priapulid worms or annelids are examples of extremely elongated, tubular animals. In saccorhytids, the antero-posterior elongation is present (as it is in the vast majority of bilaterians) but extremely reduced, *Saccorhytus* and *Beretella* having a sac-like or beret-shape, respectively. That such forms may have derived from elongated, tubular ancestors (e.g. comparable with present-day priapulid worms) would require major anatomical transformations that have no equivalent among modern animals. We agree that further speculation about the nature of these transformations is unnecessary and should be deleted simply because the nature of these ancestors is purely hypothetical. We also agree that the loss of anus and the extreme simplification of the digestive system is common among extant bilaterians. In Figure 4b, the hypothetical pre-ecdysozoan animal is slightly elongated (along its antero-posterior axis) but in no way comparable with a very elongated and cylindrical ecdysozoan worm (e.g. extant or extinct priapulid).

Therefore, I suggest to leave the evolutionary scenario more open. Supporting Saccorhytida as a true group at the early steps of Ecdysozoa evolution is important and demonstrates that animal body plans are more plastic than previously appreciated. However, with the current data, it is unlikely that Saccorhytida represents the ancestral state for Ecdysozoa (as the authors admit), and a vermiform nature is not ruled out (and even likely) in this animal group. Suggesting that the ancestral Ecdysozoan might have been small and meiobenthic is perhaps more interesting and supported by the current data (phylogeny and outgroup comparison with Spiralia).

We agree to leave the evolutionary scenario more open, especially the evolutionary process that gave rise to Saccorhytida. Again, we know nothing about the morphology of the ancestral ecdysozoan (typically the degree of body elongation, whether it had a differentiated introvert or not, whether it had a through gut or not). In Fig.4, the ancestral ecdysozoan is supposed to have evolved from a soft-bodied epibenthic animal through key innovations such as the secretion of a cuticle and ecdysis. It is a hypothesis that needs to be tested by further studies and fossil discoveries. Speculations concerning the process through which saccorhytids may have arisen have been deleted.

**Reviewer 2:**
Weaknesses:The preservations of the specimens, in particular on the putative ventral side, are not good, and the interpretation of the anatomical features needs to be tested with additional specimens in the future. The monophyly of Cycloneuralia (Nematoida + Scalidophora) was not necessarily well-supported by cladistic analyses, and the evolutionary scenario (Figure 4) also needs to be tested in future works.

Yes, we agree that the animal described in our manuscrip remains enigmatic (e.g. the natures of its internal organs, its lifestyle, etc..). Whereas the dorsal side of the animal is well documented (consistent pattern of pointed sclerites), uncertainties remain concerning its ventral anatomy (typically the mouth location and shape). Additional better-preserved specimens will hopefully provide the missing information. Concerning Cycloneuralia, their monophyly is generally better supported by analyses based on morphological characters than in molecular phylogenies.

**Reviewer 3:**
Weaknesses:I, as a paleontology non-expert, experienced several difficulties in reading the manuscript. This should be taken into consideration when assuming a wide range of readers including non-experts.

We have ensured that the text is comprehensible to biologists. The main results are summarized in relatively simple diagrams (e.g. Fig. 4) that can be understood by non-specialized readers. We are aware that technical descriptive terms may appear obscure to non-specialists. We can hardly avoid them in the descriptive parts. However, our figures (e.g. SEM images and 3D-reconstruction) are clear enough to give the reader a clear idea of the morphology of *Beretella*.

**Recommendations for the authors:**

All three reviewers appreciate the discovery and found the merit of publishing this manuscript. They also raised some concerns about the data presentation. The authors are requested to perform no additional analysis but to go through all the reviewer comments and rebut or intake them in revising the manuscript.

**Reviewer 1:**
- Line 41: comma after "ecdysozans".

OK, done.

- Formatting style: add a space before references.

OK, done.

- Line 169: B. spinosa in italics

OK, done.

- Line 157: could the "relatively large opening" in the flattened ventral side of a mouth (even when altered by the fossilisation process)?

Most bilaterians have a mouth. There is no opening on the relatively well-preserved dorsal side of *Beretella*, that could be interpreted as a mouth. In contrast the flattened ventral side often show a depressed area that could potentially bear a mouth. This ventral area is often pushed in and poorly preserved. The cuticle of this ventral side might have been relatively thinner, perhaps more flexible than that of the dorsal one (with strong sclerites). These differences might explain why the possible oral area is poorly preserved.

- Line 178: "position of the mouth"

OK, done.

- Line 219: "These sclerites, unknown..."

OK, done.

- Line 282: update reference formatting

OK, done.

- Line 298: remove reference to Supplementary Table 4, as it does not refer to the possible vermiform nature of the last common ecdysozoan ancestor?

OK, done.

- Figure 4a: change "paired legs" for "paired appendages"?

OK, done.

- Supplementary Table 4: For TGE and Introvert, the state 0 (absent) should be in bold and underlined (as it is the most likely state).

OK, done.

**Reviewer 2:**
Line 25: "from the early Cambrian" should be changed into "from the lower Cambrian"

OK, done.

Line 126: The range of maximum length should be reported in µm (rather than mm) just like those of maximum width and height.

OK, done.

Lines 191-192: Please recheck the figure panels of Saccorhytus (Supplementary Figure 4c) and scalidophoran worm (Supplementary Figure 4d). Perhaps, the former should refer to Figure 4d, and the latter to Figure 4c?

OK, done.

Lines 239 and 241: "1" and "2" appear to stand for citations (the other journal style), but I am not certain what they are.

To avoid confusing, we replace ‘1’ and ‘2’ by ‘i’ and ‘ii’.

Figures 3d and 4a: "Cycloneuralia" should be included in the phylogenetic trees.

OK, done.

Figure 3: The caption for the panel d is redundant. It should be changed into, for example, "Phylogenetic tree obtained from cladistic analyses using maximum likelihood (IQTREE)."

OK, done.

Supplementary Figures 6-9: In the captions, more detailed explanations of the results (for example, "50% majority rule consensus of XXX trees" and "strict consensus of all 4 most-parsimonious trees") should be provided.

OK, done.

Supplementary Figures 8 and 9: The caption explains that Cycloneuralia is resolved as a paraphyletic group, but it is not certain because Nematoida, Scalidophora, and Panarthropoda are resolved in a polytomy.

We changed the sentence into:

“Note that Cycloneuralia does not appear as a monophyletic clade”

**Reviewer 3:**
Line 25 'tiny' - I suggest giving an absolute measure of the size.

We add ‘maximal length 3 mm’.

Line 29 'both forms' - This is hard to follow by a non-expert. Can this be replaced with 'fossil species'?

OK, done.

Line 32 'dead-end' - Is this word necessary? I suggest to skip this word, as it is obvious that this lineage is extinct.

OK, done.

Lines 80, 94, and 172 'Remarks' - I, as a palaeontology non-expert, cannot get this manuscript structure with a repetition of this same section title.

Our systematic descriptions follow the standard rules in palaeontology.

Line 119 - I could not get what this 'Member 5' that was not introduced earlier means.

In Stratigraphy, ‘member’ is a lithostratigraphic subdivision (a Formation is usually subdivided into several Members).

Lines 104, 105, 417, ... - The name of the organization or database hosting these IDs (CUB.... and ELIXX....) should also be supplied.

OK, done.

Lines 341 and 361 - These two Figures (Figures 1 and 2) have the same caption (with an addition to the one for Figure 1). There should be a distinction based on what is presented in each figure.

We corrected the caption of Figure 2 and wrote the following: ‘*Beretella spinosa* gen. et sp. nov.’.

Line 362-367 - There is no guide about what the individual figure panels (e.g., Figure 2g, 2h, and 2i) show in detail. This guide should be supplied. This also applies to Figure 3a-c - are they anterolateral (a), dorsal (b), and posterolateral (c) views? It is better to write clearly in this way.

OK, done.

Figure 3d - The color contrast is not sufficient, and this figure does not look reader-friendly. Plus, the division into Cycloneuralia and Panarthropoda is indicated above the tree, but it is not clear what range of lineages these clades include. For example, is Pliciloricidae included in Cycloneuralia? Also, is Collinsium included in Panarthropoda? This figure looks quite unreliable, and it should be easy to fix.

OK, done.

Line 277 legend of Figure 3 - Including the parenthesis only with the program name (IQTREE) is not useful at all. Isn't it enough to describe it in Methods?

OK, done. We remove (IQTREE).

Line 380 legend of Figure 3 - I could not get where 'thicker bars' are.

Known fossil record indicated by thicker vertical bars. We added “vertical”.

Line 453 - Give full names of the methods, maximum parsimony, and maximum-likelihood.

OK, done.

Line 489 - State clearly what 'the recent paper' means.

Replace ‘recent’ by ‘present’.